# Optimal visual search based on a model of target detectability in natural images

**Shima Rashidi,** * **Krista A. Ehinger, Andrew Turpin, Lars Kulik**
School of Computing and Information Systems
The University of Melbourne, VIC 3010, Australia
rashidis@student.unimelb.edu.au, {kehinger, aturpin, lkulik}@unimelb.edu.au

## Abstract

To analyse visual systems, the concept of an ideal observer promises an optimal response for a given task. Bayesian ideal observers can provide optimal responses under uncertainty, if they are given the true distributions as input. In visual search tasks, prior studies have used signal to noise ratio (SNR) or psychophysics experiments to set the distributional parameters for simple targets on backgrounds with known patterns, however these methods do not easily translate to complex targets on natural scenes. Here, we develop a model of target detectability in natural images to estimate the parameters of target-present and target-absent distributions for a visual search task. We present a novel approach for approximating the foveated detectability of a known target in natural backgrounds based on biological aspects of human visual system. Our model considers both the uncertainty about target position and the visual system's variability due to its reduced performance in the periphery compared to the fovea. Our automated prediction algorithm uses trained logistic regression as a post processing phase of a pre-trained deep neural network. Eye tracking data from 12 observers detecting targets on natural image backgrounds are used as ground truth to tune foveation parameters and evaluate the model, using cross-validation. Finally, the model of target detectability is used in a Bayesian ideal observer model of visual search, and compared to human search performance.

## 1 Introduction

Humans have evolved a foveated visual system which, instead of processing an entire view with uniform resolution, receives higher spatial detail in the centre of the visual field (the fovea). Thus, in a visual search, the eyes make a series of movements to direct the high-resolution fovea to different parts of a scene in order to find the target [1]. This process is constrained by limitations of the visual system, namely lower acuity and degraded ability to resolve feature locations in the visual periphery, resulting in target location uncertainty [2; 3; 4]. However, most models of human visual search and attention are based on the concept of saliency [5; 6]; they do not model target detectability and are not able to predict optimal eye movement sequences. Similarly, most computer vision object detectors are based on discriminative models which do not predict target detection uncertainty[7].

A Bayesian ideal observer makes optimal decisions under uncertainty. Modelling a visual search task as a Bayesian ideal observer problem [8], we assume that the visual system computes a series of optimal eye movements that reduces the uncertainty of target location. However, this assumes that the detectability of the target is known across the visual field. This process has been modeled for simple targets in noise images, and has been shown to predict human eye movements in these

search tasks [8; 9; 10]. Various approaches have been proposed for calculating the detectability of targets in the visual field, often based on signal detection theory or estimated from psychophysical experiments [11; 12; 13; 14]. However, there has been little work on automatically estimating the detectability for object targets in *natural scenes*.

In this paper, we present a novel approach for calculating the foveated detectability of an object target on natural background images, which can be fed to a Bayesian ideal observer in a visual search task. Our approach is based on signal detection theory and uses the extracted features of *target-absent* and *target-present* images to estimate the parameters of the probability distributions. Given the distributions, target detectability is estimated as a measure of the distributions' discriminability. Our method uses a logistic regression classifier as a post processing phase of a pre-trained deep neural network in a pipeline for parameter estimation. Our model is based on biological aspects of the human visual system and considers both the uncertainty of the target position and the visual system's increased uncertainty in the periphery due to its reduced performance. To model the detectability of targets across different eccentricities, we use a model of feature pooling similar to [15]. We use human detection performance to calibrate the rate at which detectability changes with eccentricity in the model, and compare the resulting ideal observer model of visual search to human search performance, in a cross-validation design.

## 2 Related work

In recent years, visual search and fixation prediction models have been widely studied as an approach to better understand human visual attention and perception [16; 17]. Early works in this area were proposed as extensions of existing saliency maps. Saliency maps model the important locations in a visual scene which should be processed or searched [18]. A saliency map can be treated as a probability map of potential target locations and searched by choosing the next fixation based on next highest probability [19; 20]. However, these models do not model target detectability and are not able to predict optimal eye movement sequences.

Other visual search models are based on biological aspects of the human visual system, and their predictions match closely with human eye movements. The authors in [9] proposed that an ideal observer, in an effort to find the target, directs their gaze to locations in the scene which will reduce uncertainty about target location. Statistics of human eye movements have been shown to match the ideal observer model, assuming that the signal to noise ratio (SNR) of a target to a background (detectability of the target) is known at all positions in the visual field [9]. Similarly to [9], which considers search for sine wave gratings on $1/f$ noise, most previous work uses simple targets and backgrounds for which calculating SNR is straightforward [11; 21; 13].

However, it is not easy to calculate SNR for complex targets in natural scenes, and thus there is little prior work attempting to derive models of detectability for such scenarios. Dorronsoro et al. [22] propose that the detectability of sinusoidal gratings in natural scenes for human observers can be predicted via separability along the dimensions of contrast and similarity. For the same task, Oluk and Geisler [23] considered detectability as a function of local background luminance, contrast and the background's cosine similarity to the target. However, there are a large number of features or statistics of both the background and the target which may affect detectability, and these are not all easily measurable. Other work on visual search has proposed a number of features which may affect target detectability such as edge orientations, color, numerosity, and size [24; 25]. The list of known features is not exhaustive and they interact in complex ways, thus it is very difficult to estimate the detectability of targets on natural images, where the target and background may be similar to, or different, from each other on a large number of feature dimensions. This observation leads us to try the deep neural approach outlined in the next section, which can extract high level features.

Another aspect of the human visual system which should be considered in models of visual search is the effect of target eccentricity (distance from the fovea on the retina) on detectability. The first models which considered eccentricity assumed that targets were only visible within a specific radius around the fovea, and invisible outside this range [26]. Later models such as the Area Activation Model treated the detectability function as a two-dimensional Gaussian with the mean at the fovea and decreasing with eccentricity [27]. Other works [28; 29; 30] model the effect of retinal eccentricity as a loss of spatial resolution and assume that more eccentric targets are less detectable because they are blurred. However, this is not an accurate model of the fall-off in detectability over eccentricity,

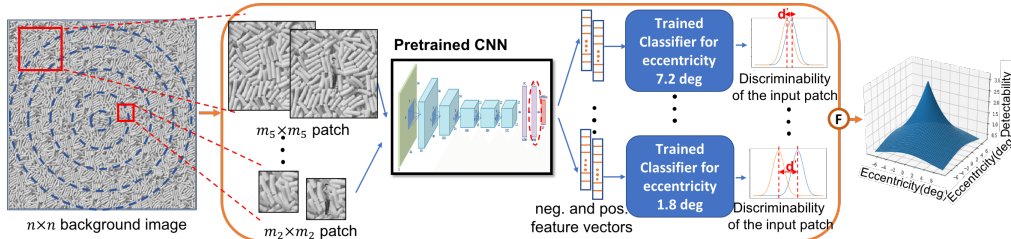

Figure 1: The overall structure of our proposed pipeline for computing a detectability map for a known object target on a natural background.

which seems to involve feature compression or feature "crowding", and not just simple image-level blur [31]. In this paper we use a model of feature pooling similar to [15; 32; 33].

## 3  Methods

In this section we describe our method for computing detectability of a given target on any natural background; the ideal observer model that we use for later comparison to human visual search [9]; and the psychophysics experiment that we used to establish ground truth for evaluating the models.

### 3.1  Proposed model

The architecture of our proposed model is presented in Figure 1. The input to the model is a gray-scale natural background with size $n \times n$ and the output is a detectability map which models an observer's ability to discriminate the known target from the given background. The model has three main components: a Convolutional Neural Network (CNN) for feature extraction; trained logistic regression classifiers for determining the probability of target presence/absence at various eccentricities; and a function to combine all of the outputs into a detectability map which also tunes the parameters of the model to match human detectabilities in this study.

Given that the human visual system has lower spatial resolution at higher eccentricities in the visual field, it is expected that the probability of target being detected correctly will decrease as the target occurs at farther eccentricities. Thus, our method for computing the detectability map takes two main factors into account: i) the discriminability of the feature distributions of the target from the given target and background; and ii) the target's distance from the center of fixation.

#### 3.1.1  Calculating distribution discriminability of a known target from a natural background

To calculate the detectability of an object target in a textured natural scene, we approximate the discriminability of target-present and target-absent distributions [34; 35]. For this purpose, we assume that the model uses a sub-optimal target-matched template. Assuming the internal noise and uncertainty of the model as a Gaussian noise, we can write the template response as a random value from standard normal distributions with means $-d'/2$ if target-absent and $+d'/2$ if target-present [12]. In this case, $d'$ is interpreted as the discriminability of the distributions, which gives us an estimate of how well separated the two distributions are from each other. $d'$ can be calculated using probabilities of hits and false alarms of the model [36]. We calculate these two probabilities using a classifier which is trained to classify patches as target-present or target-absent.

As shown in Figure 1, the input of the system is any image with size $n \times n$. In potential fixation locations, patches of the image with size $m_i \times m_i$ ($m_i < n, i \in \{1,..5\}$ which represents the five sample eccentricities used in the paper) are extracted from the image and fed to a pre-trained convolutional neural network for feature extraction. We use CNNs for feature extraction because the hierarchical feature representation they learn is analogous to the feed-forward processing in the human visual system and their representations are more similar to human visual representations than hand-tuned visual features [37; 38]. We extract features from the last fully connected layer of the CNN (before the classification layer) to use as the input to our classifier. We use pre-trained weights (pre-trained on Image-Net [39]) instead of retraining a CNN to detect the target so that the model

can be easily generalised to any target, and also to reduce the certainty of the model so that it better resembles a human observer.

The classifier is trained on the texture dataset of Cimpoi et al. [40], using patches of different backgrounds with and without the target; the dataset consists of 5640 texture images from 47 different texture categories. In this study the target is a $40 \times 40$ pixel grey-scale image of a person with a standing pose (downloaded from a commercial site *cutcaster.com*), which can be seen in the $m_i \times m_i$ patches in Figure 1. The training process is supervised; the input is the CNN feature vector for the patch and the label 1 or 0 to indicate the presence or absence of the target.

The test backgrounds which were used in the human experiments were not included in this training set, and they were taken from a different dataset ([41]). The reason for this is to show that our model is not overfit to the train dataset and can generalize to new datasets.

In order to measure the target's discriminability (denoted as $d'$) against a particular background, we present the classifier with both the background patch with no target, denoted as D (distractor), and a target-present version of the same patch, denoted as T (target). Choosing a probabilistic classifier (logistic regression) for this step, the labels are returned with a value which is monotonically related to probability (using the logistic function which normalized among the two classes so their summation equals one); the probability of the D patch being predicted as distractor denoted as $P(t|D)$ and probability of the T patch being predicted as target denoted as $P(t|T)$. In binary classifications, these two probabilities can be associated with probability of false alarm $P_f$ and hit $P_h$ respectively. These two values correspond with the area under the likelihood curves with variance of 1 and mean of $-d'/2$ [36]. Thus, $d'$ can be calculated by solving the system,

$$
\begin{cases}
P(t|T) = \phi \left( \frac{d'}{2} - c \right) \\
P(t|D) = \phi \left( \frac{-d'}{2} - c \right)
\end{cases}, \tag{1}
$$

where $\phi$ is the standard normal integral function (cumulative Gaussian) and $c$ is some decision criterion. Solving Equation 1 for $d'$ we get $d' = \phi^{-1}(P_h) - \phi^{-1}(P_f)$. Calculating $d'$ for every patch of the image, we get a detectability map as the output of the algorithm (as shown in Figure 1).

### 3.1.2 Modelling eccentricity

The intuition behind our modelling of eccentricity is based on previous work on peripheral visual processing. It has been shown that in the center of the visual field (fovea), neurons have smaller receptive fields and are closer together [42; 43]. Farther out in the visual periphery, the number of neurons decreases and their receptive fields increase in size. This means that in the periphery, each neuron covers a greater portion of the visual scene and the features are processed in less detail due to the lower density of neurons. We have modelled this property of the visual system by increasing the patch size around the target when it appears at greater eccentricity. To approximate the detectability function, we sample at five eccentricities: 0, 1.8, 3.6, 5.4, and 7.2 degrees from fixation. At each eccentricity, the classifier is trained with its corresponding patch size (see results section); so we have different trained classifiers for different eccentricities.

The detectability of the target is calculated for each of the five sample eccentricities based on these five classifiers (Figure 2). To obtain a continuous map of detectability, the five points are fit to a log-linear model $d'(r) = \alpha e^{-\beta r}$; where $r$ is the eccentricity in degree of visual angle, $\alpha$ is a constant representing the foveal threshold of detectability, and $\beta$ is the slope of the log function which controls the decrease in detectability as a function of eccentricity, which is different for each background. This function has been used to model detectability reduction as a function of eccentricity in previous studies [44]. One sample of the detectability map is shown in Figure 1 as the output of the model.

### 3.2 Visual searcher as a Bayesian ideal observer

Using the detectability maps from the model described in the previous section, we implement an ideal observer model of visual search, using the approach of Najemnik and Geisler [9].

We assume the target can appear at any of $K$ positions in the image. The observer makes a series of fixations $f \in \{1, .., F\}$ while searching for the target. On each fixation, the observer computes the probability of the target being in each potential target location ($k \in \{1, .., K\}$) while fixation is at

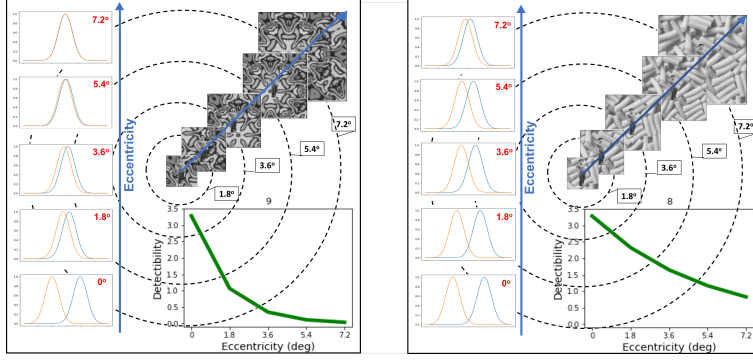

Figure 2: The probability distributions of target-absent and target-present patches become less discriminable (presented with orange and blue graphs respectively at the panel on the left side of each box) as the patches sizes increase at the five sample eccentricities (0,1.8,3.6,5.4,7.2 degrees of visual angel shown with dashed lines). With the taken approach detectability decreases as the eccentricity increases (presented as the graph at the bottom right of each box); note that the fall-off rate of detectability is different between the two backgrounds.

location $l(f)$; this probability is integrated over all fixations up until the current fixation $F$. Using Bayes' rule, the probability of $k$ being the target location as of fixation $F$ is calculated as in 2 [12]:

$$P_k(F) = \frac{prior_{k,l(f)} \times \exp\left(\sum_{f=1}^{F} d'^2_{k,l(f)} W_{k,l(f)}\right)}{\sum_{j=1}^{K} prior_{j,l(f)} \times \exp\left(\sum_{f=1}^{F} d'^2_{j,l(f)} W_{j,l(f)}\right)} \tag{2}$$

where $d'_{k,l(f)}$ is the detectability measure of location $k$ while fixation is at $l(f)$ and $W_k$ is a template response indicating the true location of the target. $d'_{k,l(f)}$ of $k$ changes with each fixation based on the fixation location, while $W_k$ remains the same throughout all fixations. The template response $W_{k,l(f)}$ is a random value derived from Gaussian distributions with mean $0.5$ at target-present locations or $-0.5$ at target-absent locations and variance $1/d'^2_{k,l(f)}$ at all locations. Given $P_k(F)$, the next fixation location, $l(f+1)$, is chosen to maximize information gain using equation 3 [9].

$$l(f+1) = \underset{l(f+1)}{\mathrm{argmax}} \left(\sum_{k=1}^{K} P_k(F) d'^2_{k,l(f+1)}\right) \tag{3}$$

This process continues until a termination criterion is met; e.g., the probability of location $i$ containing target exceeds a threshold value. The threshold value is arbitrarily set to 0.99 to model observer's confidence when target is found. We obtained similar results for threshold values in the range 0.90-0.99. The model does not require a separate inhibition of return mechanism because fixating a location reduces uncertainty at that location. If a location is fixated and the target isn't found, the model's estimate of the target probability at that location drops. This decreases likelihood of a return saccade.

### 3.3 Human data

The psychophysical experiments included two tasks: a detection task, in which the target was presented on a variety of backgrounds at various eccentricities while the observer fixated at the center of the screen; and a visual search task in which observers were asked to find the target on various backgrounds, and could move their gaze naturally.

These experiments were based on those used by Najmenik and Geisler [8], but used object targets and natural image backgrounds in place of $1/f$ noise. The backgrounds were 18 images from the ETH dataset [41]. The ETH dataset consists of 21,302 texture samples from different categories. To choose the 18 backgrounds, we ran the entire dataset through our pipeline to find images with a wide range of apparent difficulty, resulting in the 18 backgrounds shown in Figure 3 with a variety of low, medium and high detectabilities, and with a variety of image content and patterns.

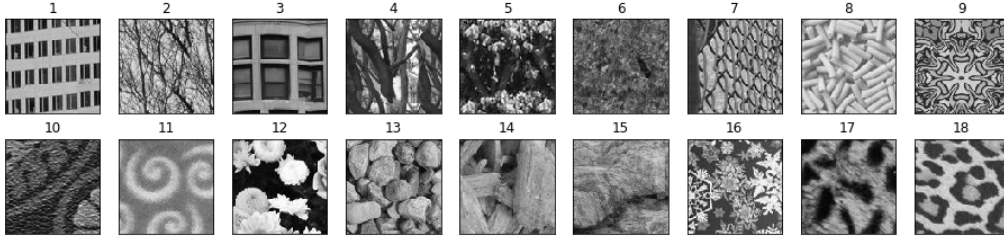

Figure 3: The used background images for the experiments chosen from the ETH dataset [41].

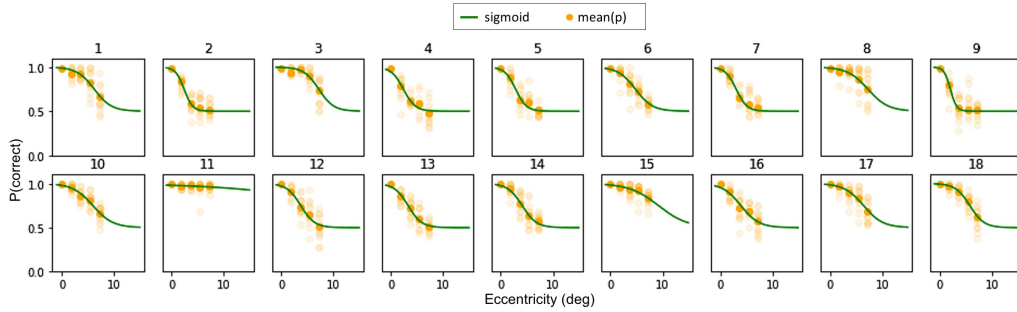

Figure 4: Probability of correct detection at each location for the 18 backgrounds (orange points). The green curves are sigmoid fits to the means of the shown data-points.

Textured backgrounds were used instead of natural scenes because they are relatively homogeneous, so we can expect similar search performance no matter where the target is located in the image; and because they provide no contextual priors for target location, which may bias search paths [45]. However, the model described here can be extended to natural scenes – a separate detectability can be computed for each region of a heterogeneous scene, and expected target locations can be incorporated as a prior in (Equation 2).

Each image was presented at a resolution of $666 \times 666$ pixels and subtended 15 degrees of visual angle. The background was randomly cropped from a larger field of texture created through texture quilting [46]. Images were presented on a $1680 \times 1050$ pixel LCD screen, on background set to the mean luminance of the image. The screen width and distance were 48cm and 67cm respectively.

### 3.3.1 Detection

In the detection task, 12 participants (age range 20 - 40) judged if the target occurred in the first or second frame in a two-alternative forced choice (2AFC) paradigm. The experimental procedure received ethics approval from University of Melbourne Human Research Ethics Committee (ID: 1955695). Observers were required to fixate at the center of the display and maintain fixation throughout each trial. Their fixation was constantly monitored with an eye tracker (Gazepoint Version 2.10.0). If the eye position is moved 0.9 degree from the center of the display during the stimulus presentation, the trial was discarded and repeated at the end of the session. Each session consisted of 32 repetitions of 18 backgrounds with the target occurring at the same location throughout the session. Targets were located at one of four positions spaced 1.8 degrees apart on a 45 degree radial from fixation. To reduce the length of the experiment, we assumed that human foveal detection accuracy is essentially 100% for these backgrounds and did not test detectability at fixation. Participants were informed of the target location at the start of each session and were also post-cued by a circle which appeared around the target for 0.2ms at the end of each trial. The whole experiment consisted of four sessions (four eccentricities) and each session was around 20 minutes. Each session was preceded by a short practice session of 20 trials with a sample background, to familiarize the participant with the target appearance, location, and experiment process.

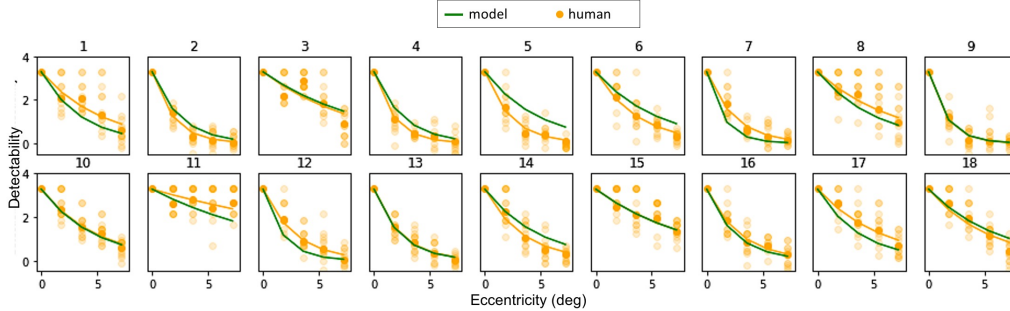

Figure 5: Detectability estimates from the model (green curve) and from human observers (orange curve) for the 18 backgrounds. Orange data-point show the range of human detectabilities.

The averaged probability of correct detection across the 12 participants for each background is shown in Figure 4. The probability of correct detection $p_{correct}(r)$ for each background at eccentricity $r$ is fit to a sigmoid function as in $p_{correct}(r) = \frac{0.5}{1+\exp((r-\mu)/\sigma)} + 0.5$ where $\mu$ and $\sigma$ are constants that indicate inflection point and slope of the sigmoid. For a 2AFC, these probabilities must lie in the range 0.5 to 1. As seen in Figure 4, the sigmoid functions vary over the different backgrounds. The probability functions show some variety among observers, but they are consistent with previously reported measurements [47; 48; 12] in terms of having the highest probability of detection at fovea ($r = 0$) and decreasing probability as a function of eccentricity. Background 9, with the lowest $\mu$ and $\sigma$, and Background 11 with highest $\mu$ and $\sigma$ have the highest and lowest detectabilities, respectively. At 1.8 degrees of visual angle, the probability of correctly detecting the target in Background 9 is high, but it quickly falls to chance at visual angles further than 4 degrees. In contrast, the probability of detecting the target on Background 11 is high at all tested eccentricities.

### 3.3.2 Visual search

Due to COVID-19 restrictions, there were only two participants from the detectability task who were also able to participate in the visual search task. In the visual search task, observers were shown the 18 backgrounds with the target pasted randomly in one of 84 locations (chose to uniformly tile a 15-degree circle within the background [12]). Observers were asked to find the target as soon as possible and press a key when found. Fixations were recorded and the final fixation before pressing the key was checked against the target location. If the fixation was closer to the target location than to any other potential target location, the trial was considered a correct detection, otherwise it was discarded. If the target was not found within 20 seconds, the trial terminated automatically; these timed-out trials were also discarded. The number of fixations on each search trial were compared to the predictions of the ideal observer model.

## 4  Results

We evaluate the performance of our model using different feature representation pipelines to compute the detectability of the target against each background. The input of the pipeline for the fovea are $224 \times 224$ pixel target-absent and target-present patches. In target-present images, the target, sized $212 \times 212$ pixels, is pasted at the center. 100 random crops of the background are used and the results are averaged to compute the detectability. We ran the experiments on a high performance computing server with a 16GB V100 GPU, hosted by University of Melbourne [49].

At further eccentricities, the patch size is scaled to $[1.4, 1.8, 2.2, 2.4]$ of its original size at the sample locations 1.8, 3.6, 5.4, and 7.2 degrees from the fovea. As this scaling is not necessarily equivalent to the feature pooling in the human visual system, we fit the detectabilities to the human detectabilities with an appropriate scaling and translation for each background. The formula that we use is

$$w_i = \underset{w_i}{\operatorname{argmin}}(w_i(d_{M,i} - b) - d_H)^2, i \in \{2, .., 5\} \tag{4}$$

in which $i$ iterates over the 5 eccentricities, $b$ and $w_i$ are the translation and scaling parameters, and $d_M$ and $d_H$ are the model and human detectabilities respectively. The translation coefficient is

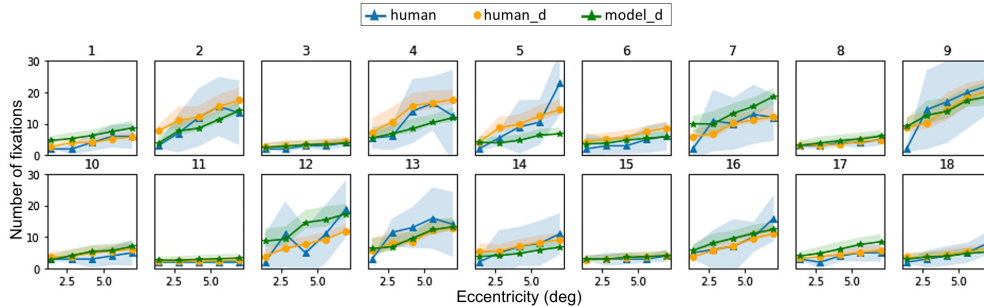

Figure 6: Number of fixations in the visual search task for the 18 backgrounds. Blue curves represent the performance of human observers; orange and green curves represent the Bayesian ideal observer based on human or model detectabilities, respectively.

calculated as $b = d_{M,1} - 3.28$ in which $d_{M,1}$ is the detectability of the model in the fovea and 3.28 is the human detectability at the center for any background. The scaling parameter $w_i$ is chosen for each background by finding the best fit to the other backgrounds in a leave-one-out cross-validation. The results are shown in Figure 5.

Using $p_{correct}$ from the human experiments, the ground truth detectability of the target on each textured background can be calculated using $d'(r) = 2\Phi^{-1}(p_{correct}(r))$ in which $\Phi^{-1}(.)$ is the inverse cumulative standard normal distribution. The detectabilities calculated from the human experiments using the equation above, and the output of our proposed model for the four eccentricities are shown in Figure 5 using the best feature pipeline from Table 1.

The MSE of the model output compared to the human detectability for different feature representations and classifiers are shown in Table 1. In addition to features derived from CNNs, we also ran our model using a simple luminance histogram to represent the target and background, as well as a few hand-crafted features which have been proposed as analogues to the human visual system [50]. The classifiers used were logistic regression and MLP (a two-layer neural network).

| Features | MSE | SE | Features | MSE | SE |
|---|---|---|---|---|---|
| Alexnet + Log. Res. | 0.0978 | 0.0015 | luminance hist. + Log. Res. | 0.3404 | 0.0057 |
| Alexnet + MLP | 0.256 | 0.0035 | HMAX[51]+ Log. Res. | 0.224 | 0.0041 |
| VGG19 + Log. Res. | 0.257 | 0.0033 | Textons[52]+ Log. Res. | 0.2767 | 0.0036 |

Table 1: Mean squared error (MSE) and standard error of the mean (SE) of various models for predicting human detectability (see text).

As can be seen, Alexnet+logistic regression gives the best fit. Alexnet+MLP has a high confidence classifying the target-present and target-absent images, so causes high detectabilities even for backgrounds with complicated patterns. The output features of VGG19 [53], which is a 19-layer CNN, seem to perform worse than Alexnet. This might be due to VGG19's deeper structure which results in higher confidence predictions of the classifier. This causes the model to underestimate the difficulty of some backgrounds that are difficult for human observers. The results in the table confirm that for our purpose, simple features such as luminance or oriented filter banks [52] do not provide a sufficient representation of the image. More complex features are needed to produce detectability maps similar to those of humans. HMAX [51], which mimics the operation of simple and complex cells in the human visual system, is quite competitive with deeper CNNs.

Finally we use the modeled detectabilities in the Bayesian ideal observer model formulated in Equation 2 to predict the number of fixations needed to find the target on these backgrounds. Number of fixations is a measure of search performance which should depend on the detectability of the target, which in turn is a function of the background and eccentricity. The results of the visual search algorithm from this simulation compared to the actual number of human fixations from the visual search experiment are shown in Figure 6. The ideal observer model closely follows the human observers' number of fixations; the results from the full model (using model $d'$ from the feature

pipeline) and an "oracle" model (which uses the human $d'$ from the detectability task) are similar. However, human observers perform worse than the ideal observers in some cases, which may be due to environmental and internal distractions.

## 5    Conclusion

In this paper, we propose a method for computing a foveated model of detectability for object targets in natural backgrounds that closely mimics human performance. We use these detectabilities in a Bayesian ideal observer model of visual search based on [12] and are able to simulate human search performance. A comparison of different feature pipelines confirms that more complex visual features, such as those computed by CNNs, are needed to estimate the detectability of objects in natural images, though deeper CNN architectures are not necessarily better at producing human-like representations. This work adds to a growing body of literature showing the potential of CNNs to serve as an approximation of the feed-forward visual processing pipeline in humans, enabling the development of more sophisticated models of visual attention and fixation control.

## Acknowledgments and Disclosure of Funding

This research was undertaken using the LIEF HPC-GPGPU Facility hosted at the University of Melbourne. This Facility was established with the assistance of ARC LIEF Grant LE170100200 [49].

## Broader Impact

The work presented in this paper can be used for attention prediction systems. Such systems can be beneficial for applications such as augmented reality driving aids [54; 55]. A foveated detectability model could effectively model a driver's visual system and quantitatively measure the detectability of driving hazards based on the driver's current gaze. This system could direct attention to locations with high uncertainty to reduce the possibility of a hazard being missed that is not in the driver's fovea. However, attention prediction can also be abused for negative outcomes such as manipulating attention for advertising. The proposed system is not intended for high risk applications or if applied, should not have a critical role in them, so the consequences of failure of the system should not be severely destructive. The proposed model is not leveraging any dataset biases, but the model of detectability requires training on a large set of backgrounds, so an unrepresentative selection of backgrounds could produce a wrong model and could potentially limit the results.

## Footnotes

*The code to reproduce the results of the paper can be found at `https://github.com/rashidis/bio_based_detectability`

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
