[Reviews · NeurIPS 2020]

Review 1

Summary and Contributions: This paper presents a visual search model that is based on latent features of a CNN tuned to perform target detectability as a function of retinal eccentricity (computed via different patch sizes as shown in Figure 1). The model is the fitted with human data to real target detectability experiments in a forced fixation experiment, and later deployed and compared with real human observers engaging in a visual search task of a target cluttered with background texture. -- Post Rebuttal -- : I think authors have done a good job in the rebuttal, and after reading other reviewers thoughts and discussions, I have decided not to change my score, and still leaning towards accepting this paper.

Strengths: * The paper dramatically extends the work of Najemnik & Geisler that is limited to a gabor in 1/f noise. * The paper has high quality psychophysics exploration methods and proper use of signal detection theory to compute target detectability. (a good thing! This is not "yet another CNN over-fitting visual search model) * The paper is easy to read, the ideas are understandable and principled, and the results, claims and tone sound reproducible. * High amount of behavioural data for the first experiment (12 observers). * Ultimately, I am leaning slightly more towards accept (likely as a poster) vs reject because this paper is cross-disciplinary and could gather crowds from vision science.

Weaknesses: * Please See Correctness. * Missing References (see Relation to prior work section) * Limited human observers (2) for the "experimental crux" of the paper which is the visual search task. * Can Figure 6 be numerically quantified to better understand the quality of the computational modelling of the visual search task?* Also on this note, it is possible that a better control (analogous to computing a noise ceiling) to actual human eye-movements form the 2 observers is not only the human and model detectability, but rather a randomly saccading observer.

Correctness: I am a bit on the fence with this paper. On one end the psychophysics is strong (despite having a weird motivation where the proxy for eccentricity dependence is randomly picking different sized patches -- though this problem is overcome with the curve fitting later presented in Section 3 and early Section 4). On the other hand, is this really an end-to-end CNN-based model of visual search? This model does not receive an input image I, pick a point of fixation (x,y) and somehow implement foveated processing to later compute if the target is presented/absent (and recursively iterate); also the search procedure does not seem to implement operations such as inhibition of return. The method is a bit convoluted, but from a psychophysical perspective it seems to deliver via results in Table 1, but Figure 6 shows a non-ideal evaluation metric. Perhaps MSE of the (x,y) coordinates of the searcher model would be better than number of eye-movements as the pattern of saccades could be different despite converging to the same number before finding the target. In any case, I am curious to see both what other reviewers think and if the authors can address some of my concerns that I may have missed when reviewing this paper.

Clarity: Yes, although figure 4,5,6 could use some slight brushing to highlight the axis better.

Relation to Prior Work: Important: ** Line 38: On estimating target d' via Foveated Feature Pooling and as a function of eccentricity, see Deza & Eckstein NeurIPS, 2016, and Deza et al. ACM CHI, 2017. ** Line 48: On Visual Metermism via local texture computation in the periphery: See Wallis et al., 2019. eLife. (Scene content is more important than Bouma's Law) and Deza et al. 2019. ICLR. (Foveated Style Transfer) * Section 3.2: Visual Search as a Bayesian Ideal Observer: Eckstein et al, 2006, Torralba et al. 2006. *Also the conclusion: Line 288: Akbas & Eckstein 2017. PLOS Computational Biology. (Foveated Object Detector) Broader level citations: **Also see additional Feedback. ** Line 53: Eckstein, 2011 (Visual Search). ** Line 90: Rosenholtz 2016 (Peripheral Vision).

Reproducibility: Yes

Additional Feedback: At some point the Side-Eye model from Fridman et al. 2017 is discussed and said "used as an inspiration for our model", but I am not sure how this is the case given that the SideEye model is a generative model and renders what looks like a visual metamer (Freeman & Simoncelli; Rosenholtz (TTM), 2011) -- while the model proposed in the paper is using varying sized patches to compute an estimate of d' as a function of "retinal eccentricity", although the eccentricity is implicitly modelled through the size of the patch vs an actual spatially adaptive model like the one proposed in Poggio et al. 2014, Chen et al. 2017, Han et al. 2020. Typo? Line 269: 2-layer, .. hmm should this be 2-stage? I don't think AlexNet is 2 layer unless I am missing something. Typo? Line 278: Section 2. The conclusion reads too short, I wish there would have been a more thorough discussion of the results. I understand that the covid situation limited the visual search experiment, but what other analysis could be done to highlight the strengths of the paper? Maybe re-writing the last page such that the paper only encompasses a single forced fixation task is an option with a more thorough discussion? This is up to the authors to decide!


Review 2

Summary and Contributions: Presents a method for estimating the discriminability of a target from a textured background, and provides a model for target detectability as a function of eccentricity for human vision.

Strengths: The idea of considering detectability and uncertainty is an important one, which has not been addressed enough in the image saliency literature. (but the paper does not explicitly model uncertainty - instead it computes d-prime values). The modeling of detectability with eccentricity, via curve fitting to sampling of multiple trained estimators at different eccentricities is good.

Weaknesses: The text describing figure 1 mentions a "gray scale natural background". What is natural about a gray scale background? The image in figure 1 does not look like a natural background. It looks like a uniform synthetic texture with very little structure. In section 3.1.1.1 the paper states (line 128) that the system is trained on patches from a texture dataset. There should be a realistic model for natural background images and their features. The "known" target is fixed to be an image of a standing person. This is quite restrictive, and implies that the method would need retraining for every possible target to be detected. For a specific application (e.g. finding people in a search-and-rescue situation) this is acceptable, but for use in a general model of human perception it is not. In figure 2 it appears that the scale of the target (standing person) is quite unrealistic as compared with the scales of the textures elements. Typically in real natural images a patch containing a whole person would have the background be quite varied in structure, nothing like the uniform textures used here. The paper claims that "However, the model described here can be extended to natural scenes – a separate detectability can be computed for each region of a heterogeneous scene, and expected target locations can be incorporated as a prior in (Equation 2)." However, in natural scenes a target may be in areas where the immediate background is itself heterogenous, and this may be the most common situation. A more complex background scene model would be needed to model human performance in real imagery.

Correctness: Yes.

Clarity: Yes.

Relation to Prior Work: Navalpakkam and Itti had a paper that modeled target detection in natural scenes that built a statistical model of the visual features of target and scene clutter. This seems similar to the proposed method, and should be cited. From the paper: "The topdown component uses accumulated statistical knowledge of the visual features of the desired search target and background clutter, to optimally tune the bottom-up maps such that target detection speed is maximized" Navalpakkam, V., & Itti, L. (2006, June). An integrated model of top-down and bottom-up attention for optimizing detection speed. In 2006 IEEE Computer Society Conference on Computer Vision and Pattern Recognition (CVPR'06) (Vol. 2, pp. 2049-2056). IEEE. Itti, L., Gold, C., & Koch, C. (2001). Visual attention and target detection in cluttered natural scenes. OptEn, 40, 1784-1793. Rotman, S. R., Tidhar, G., & Kowalczyk, M. L. (1994). Clutter metrics for target detection systems. IEEE Transactions on Aerospace and Electronic Systems, 30(1), 81-91.

Reproducibility: Yes

Additional Feedback: The sentence (line 123-124) "We extract features from the last fully connected layer of the CNN (before the classification layer) to use as the input to our classifier" is circular. It should be rephrased. The caption for figure 2 mentions both "detectability" and "discriminability", but only discriminability is shown. The associated text does not provide a definition of detectability. A definition is only given for discriminability. Is it the case (as seems to be implied by its usage in section 3.1.2) that detectability is actually the same quantity as discriminability? If so, why use two different terms? If they are actually different, then a clear definition needs to be given for detectability. Why were the background patches used in the experiment (section 3.3) different than those used to train the detectability map system? The target object was not defined in the experiment (section 3.3). Was this the same standing person shape that was used in training the system? The results section 4 seems to be just for the results of the visual search task (section 3.3.2). Either put the results of the detectability task into section 4, or put the results of the search task in section 3.3. Regarding the impact statement a definite risk of this type of work is that it does not fully model realistic natural scenes, and so application in real-world situations, may result in incorrect decisions being made which could have disastrous outcomes. I think this paper is perfectly relevant for Neurips, at least based on the historical range of neurips/nips papers, especially in the early years of the conference. It is about modeling the human visual system. I think the neural network developers could benefit from thinking about detectability and eccentricity issues. I thought the rebuttal was just "OK". I don't buy the argument about d-prime and uncertainty, as d-prime folds in two factors - signal separation and noise level. d-prime could be used as a measure of uncertainty in some sense, but the exact problem being considered needs to be precisely defined. I usually think of uncertainty in a Bayesian decision theory sense, which would result in a different view of uncertainty measures than d-prime values.


Review 3

Summary and Contributions: This paper uses pre-trained deep networks, as well as a few other systems (e.g., HMAX), to extract features from images with different sizes of a target relative to the image patch size, simulating eccentricity from the fovea. The targets are placed on various backgrounds drawn from the Describable Texture Dataset (DTD). The features are used as inputs to logistic regression models, one for each eccentricity, to predict the presence or absence of the target. The resulting logistic regression models are used to compute a P(detect) and P(false_alarm), resulting in a d’ for every eccentricity and background. Then a human psychophysics detection experiment is used to compute human d’s for 18 different backgrounds (from a different dataset) and five eccentricities, starting with a foveal image. Finally, the d’s of the model are linearly transformed to fit the d’ of the humans due to uncertainty about whether the eccentricities in the model correspond directly to human eccentricities. Finally, these measurements are then incorporated in a Bayesian optimal model of visual search. The model is able to predict with very good accuracy the number of fixations required to find the target for the 18 backgrounds. After reading the rebuttal, I am satisfied with their response. As I was in favor of acceptance to begin with, my review score doesn't change. I was slightly perturbed that in the response they say they are focused on fixation order in one paragraph and explain why they can't do fixation order in another, and again, the results are about number of fixations, not the order of fixations. If they implemented their model on natural images, then they could perhaps predict fixation order, but they should not claim that for this paper.

Strengths: 1. The authors claim that this is the first model of visual search that works with natural images. The model is completely novel, as far as I know. 2. They do psychophysical measurements of target detectability on 18 different backgrounds, and use this measurement to adapt their Bayesian ideal observer model of visual search to the human data. 3. The model is able to predict the number of fixations required to find the target. 4. They compare different feature extraction methods and show that Alexnet, but not VGG16, fits the data quite well.

Weaknesses: 1. While the authors state that they measure target detectability in natural images, and the images are natural, they are textured images from a particular dataset, which consist of 18 different texture types. However, while these are “natural,” they do not correspond to, for example, images of natural scenes, which are much more complex. I think the authors should be a little bit careful about the claims, as natural images are composed of many different textures and objects in the same image. To their credit, the authors recognize this and describe how their model could be applied to natural scenes, but they don’t do that here. 2. The linear fit between the d’s in the model and the d’s in humans were fit individually to each eccentricity. It would be more convincing if the authors had used one overall fit. However, the fact remains that the Alexnet model provides a fit that is way better than the other models, despite these extra parameters. 3. The abstract states that they compare their model to human fixation patterns. This sets up the reader to expect scan paths, but all we get is a fit to number of fixations until detection. It would be best to recalibrate this claim.

Correctness: They appear to be.

Clarity: Fairly well written. There are places where not all variables in an expression are defined. For example, I assume “epsilon” stands for eccentricity in the log-linear model used to create a smooth detectability map? Minor points: lines 18-19: say fixation numbers, not patterns. line 33: of target -> of the target. line 89: use two single quotes in latex instead of the “ character for the right hand quote mark. Figure 1: this is pretty much impossible to read without blowing up the pdf. Throughout: Replace data-set with dataset (one word). line 140: Not sure what you mean here by “normalized”? line 146: by “standard normal integral function”, I assume you mean the cumulative gaussian? line 156-158: remove one of the phrases “to approximate the detectability function,” presumably the one at the end of the sentence. Line 177: Add the word “is” at the end of the line. lines 181-183: How did you determine the threshold value? line 214: five positions, I think you mean (including the foveal one). Line 229: This sentence no verb. Also no direct object. (before the comma). Line 231: remove: “which shows its high detectability.” Line 261: What are e and b here? Is the b here the same as the b in eqn 4? Shouldn’t at least one of them occur on the right hand side? I’m a little confused about how you can compute d’ from only the probability of correct detection. Line 269: Be clear that by “a two-layer neural network”, you are referring to MLP in the table. Lines 278 and 289: Is the bayesian idea observer model from reference 2 or reference 12?

Relation to Prior Work: Yes, and the reference list is extensive (52 references!).

Reproducibility: No

Additional Feedback: If the authors would make the code available (if accepted), that would enhance replicability.


Review 4

Summary and Contributions: This paper proposes a way to measure target detectability in naturalistic scenes, which is the topic that has not been well investigated in the visual search literature. The approach uses CNNs to extract features and uses signal detection theory to estimate target detectibility. The authors resolve differences in detectability across target eccentricities by increasing background size while keeping target size constant. The parameters were calibrated using human data.

Strengths: - Using CNN features is a robust and scalable approach to computing detectability. - Comparison to human data nicely validates the proposed approach.

Weaknesses: In my view, the major drawback of this work is its limited relevance to the NeurIPS community. If the goal is to be able to predict eye movements or attentional shifts, then it is unclear whether this approach is better than previously proposed approaches that may not necessarily be Bayesian-optimal, but may nonetheless work well in practice. I'm not an expert in the field of visual search, but I can't help wondering how this approach compares to the work from Bethge lab (DeepGaze), which was published back in 2015-2016. (And surely many other papers building on that work have been published since.) As presented, it seems that this work is concerned primarily with introducing CNNs to the visual search literature in a very limited artificial target detection task. I think for this work to be more relevant to NeurIPS community, the task would have to be broader (e.g., predicting eye movements) and direct comparisons to other state of the art models would have to be provided. As is, this work seems to be too specialized.

Correctness: The reasoning how the model is constructed seems correct, but due to me not being an expert in this field, I cannot evaluate whether the Bayesian idea observer formulation is correct. I was also skeptical about the poor performance of VGG-19. The big difference between AlexNet and VGG-19 in this task suggests to me that either something is wrong about inputs to VGG-19 (e.g., maybe preprocessing step was not carried out correctly?) or there is a huge variability across tasks (e.g., if the backgrounds were not textures but landscapes), in which case more testing is needed.

Clarity: Yes, though I was struggling to understand the precise goal of the study and how different component related to each other (e.g., how fitting to the data was done – it is described but I wasn't sure I really understood it).

Relation to Prior Work: Yes.

Reproducibility: Yes

Additional Feedback: I have read the rebuttal and other reviewers comments. On the one hand, I realized that my point about the relevance to NeurIPS community was poorly stated and led to unnecessary discussions about subjective beliefs. So my reservations are better stated this way: I think it would be important to show that the proposed approach works on any image (that is, the model is image-computable) or that the approach is really novel. From the rebuttal, it seems that the model might be extended to be image-computable. Whether the approach is sufficiently novel, I am unsure as I'm unfamiliar with this literature. On the other hand, I'm not very convinced that this approach was not compared to other models because those "focus more on fixation location than fixation order". Maybe, but "focus" does not mean that fixation order is definitely unavailable in these models. Also, the paper itself only reports the number of fixations, not the order, as far as I can tell, so this explanation did not seem convincing. Also, my concerns about really poor VGG performance was not addressed. When I was just starting with deep nets, I remember having all sorts of weird issues due to incorrect image preprocessing and other subtle details. I am concerned that this may also be an issue here (though this is a minor point). Overall, I increased my score from 3 to 5 to give this paper a chance if other reviewers felt strongly that this work deserves to be accepted.

[Author Response · NeurIPS 2020]

We thank the reviewers for their valuable and generally positive feedback. We are encouraged that the reviewers found our work novel in terms of "not being yet another CNN over-fitting visual search model" (R1 and R3), a significant improvement over Najemnik & Geisler (2005) (R1), supported by theory (R1) and rich in psychophysics (R1 and R3) with high amount of behavioural data (R1). We are pleased that R2 points out that considering detectability and uncertainty is an important aspect of our work that has not been addressed sufficiently in the image saliency literature. We appreciate the helpful suggestions regarding improvements to figures and wording and will incorporate these in the camera-ready version. We provide responses to the major concerns below.

**@R4: Concern about the work's relevance to the NeurIPS community** The goal of the paper is to model the process by which the human visual system makes saccades during search, based on a Bayesian ideal observer model; that seems like a suitable topic for the "Neuroscience and Cognitive Science" area. This focus is similar to previous NeurIPS papers (such as Deza & Eckstein, 2016; Yu, Hua, Samaras & Zelinsky, 2013; Smeulders & Lamme, 2009, among many, and we note that the other reviewers found the paper relevant.

**@R4, R2: Comparison to existing saliency models** We did not compare this model to existing saliency models such as DeepGaze because our goal with this approach is significantly different from those models. Deep-learning saliency models are fit to large training datasets of fixations using machine learning techniques; generally, they aren't built on models of visual processing and they focus more on fixation location than fixation order. Here, we wish to predict fixation sequences using a perceptual model of target detectability across the visual field.

**@R2,R3: modeling uncertainty** @R2: To approximate the uncertainty in a visual search task, we only need the $d'$ which is a measure of discriminability between a target-present and target-absent patch. Thus, an estimate of the distributions of the target and background is not required. @R3: The area under the curve of likelihood distributions, $p(t|T)$ and $p(t|D)$, are associated with the probabilities of hit and false alarms, and calculated from eq1. As illustrated in Figure 2, the detectability $d'$ can be calculated as the distance between the means of the two likelihood distributions.

**@R1: Is this really an end-to-end CNN-based model of visual search? Why doesn't the model implement inhibition of return?** The proposed pipeline can output the detectability map for any background and target pair. The model does not require inhibition of return because the Bayesian-update step after each fixation updates the prior via the posterior. This naturally causes the probability of target-present events to decrease for any previously-fixated locations that did not contain the target, thus reducing the likelihood of a return saccade.

**@R1,R3: Number of fixations rather than scan-path as a measure for validating the model.** Since the visual search is implemented on a statistically-stationary textured background, the scanpaths of different observers are expected to be quite different. The lengths (and by extension, number) of saccades should be similar because these depend on target discriminability. However, the directions of saccades may not be similar for all observers because one observer might make an initial saccade to the left, another to the right, etc. This makes it difficult to compare scanpaths.

**@R2: Regarding the use of textured images rather than natural scenes, the single target and the scale of the target on the backgrounds.** As mentioned in line 198 of the paper and noted by R3, the model is not limited to homogeneous textures and can easily be extended to natural scenes. However, the goal in training was to model target detectability on the widest possible range of backgrounds, not just the types of backgrounds on which a pedestrian target is most likely to appear in real-world scenes. Similarly, when choosing backgrounds for the human experiment, we chose backgrounds which exhibited a range of detectabilities in the model. We consider the scale of the background relative to the target to be irrelevant, because the goal is to model the perceptual detectability at different eccentricities – for this, it is most important to have a variety of target-background feature contrasts.

The model can be extended to natural scenes by considering detectability in small patches (size dependent on eccentricity) and computing a heterogeneous detectability map over the entire image. The training set of the current model includes fairly heterogenous large-scale textures, and many cases where the target fell on the boundary between two different-looking regions, so these cases should not be an issue for extending the model. To extend the model to different targets, it is necessary to recompute the detectability for that target, but this doesn't require retraining the CNNs, only the decision boundary between the target and the background. We believe this would be necessary for any human-like model of target detectability: detectability does not seem to be explained by low-level feature contrasts, so there is no simple function that could be computed in pixel space to predict discriminability of any target at any eccentricity on any background. The model is intended as a simulation; more testing on a broader range of stimuli and participants would be required before it could be deployed in a real-world application

**@R1, R2: addition of related work** The inspiration taken from Fridman et al. (2016) and the related Rosenholtz and Freeman work is the use of spatially larger feature-pooling regions to represent the feature compression in the visual periphery. Unlike Deza & Eckstein (2016), we compute detectability for specific targets based on the feature contrast with the background. Our signal-detection-based model is similar to Navalpakkam, V., & Itti, L. (2006) but considers more complex features to compute target detectability.

[Meta-Review · NeurIPS 2020]

This paper presents a method to measure target detectability in natural images. It provides a visual search model (based on extracted features of a pre-trained CNN) to perform target detectability as a function of retinal eccentricity for human vision. Reviewers, including myself, appreciate that this paper tackles a topic that has not been well investigated in the visual search literature. The approach is well-motivated and paper is well written, and comparison with human data is a nice validation of the approach. There were issues concerning correctness of the approach, along with minor points, but the author's rebuttal has done an adequate job in addressing the concerns and I expect to see the camera ready version of the paper incorporate improvements to at will improve the clarity of the paper (esp with regards to reviewer's main concerns) using the extra page. I think this will be a nice addition to the NeurIPS2020 conference encouraging the community to look at a fresh topic, so I'm going to recommend we accept this work as a poster.